# Exploring Energy Security and Independence for Small Energy Users: A Latvian Case Study on Unleashing Stirling Engine Potential

**Janis Kramens, Oskars Svedovs *, Amanda Sturmane, Edgars Vigants, Vladimirs Kirsanovs and Dagnija Blumberga**

Institute of Energy Systems and Environment, Riga Technical University, Āzenes Street 12/1, LV-1048 Riga, Latvia; janis.kramens@edu.rtu.lv (J.K.)

*   Correspondence: oskars.svedovs@rtu.lv

**Abstract:** Nowadays, energy systems are continuously impacted by external and internal conditions. The worldwide events of recent years have led to the need to consider not only the requirements for moving towards climate neutrality but also the security and independence of energy supply when creating new or transforming existing energy systems. The aim of this study was to answer the question of whether there is a possibility of transforming the energy supply process by promoting increased energy security and independence while not reducing energy demand, as well as creating high economic and environmental indicators. The research focuses on developing alternative scenarios for further decision-making studies by introducing modern energy technologies. Scenarios are analysed using the complex method developed, which includes three main steps: assessing the current situation, identifying available technologies, and assessing alternatives. The results suggest that Stirling's technology can provide 100% energy independence for individual energy consumers. At the same time, thanks to the combination of Stirling technology and solar technologies, there is an opportunity to reduce emissions and energy production costs, but capital investment is increasing.

**Keywords:** energy security; sustainable development; energy consumption reduction; energy technologies; Stirling engine; specific energy cost; emissions

## 1. Introduction

### 1.1. Definition and Importance of Energy Security

Energy security is defined globally as the continuous availability of affordable vital energy systems [1]. Vital energy systems are energy resources, infrastructure, and use linked together by energy flows that support different functions [2] that cover five categories of aspects: political, social, economic, technical, and environmental [3]. By simplifying the structure of energy security and combining different aspects, four main pillars are expressed: availability (geological), accessibility (geopolitical), affordability (economic), and acceptability (environmental) [2,4–7]. Pillars are linked by groups of external and internal factors, as shown in Figure 1. Interactions between these factors are a process to maintain the stable functioning of pillars, which is called sustainable development [8–11].

Jacek Strojny et al. describe the broad energy security concept issue [12]. The authors point to three main ways of energy security concepts—traditional approaches, index approaches, and "contemporary" approaches. The traditional approaches were based on the four-pillar system mentioned above. According to the authors, the use of such approaches decreases annually, according to the number of publications. Two other types of approaches are much more advanced. In similar studies, index approaches are considered a primary method that attempts to quantify selected aspects of energy security. A literature study on the subject has shown that the nature of the concept of energy security may vary depending on the potential objectives to achieve or the challenges to address. Debates about energy security are common in political and scientific fields. There is a problem in choosing

primary sectors that should prioritise resources—some see the military sector as the most important, while others see the private sector as the priority. However, the main thing that must underpin energy security concepts is the diversification of resources [12,13].

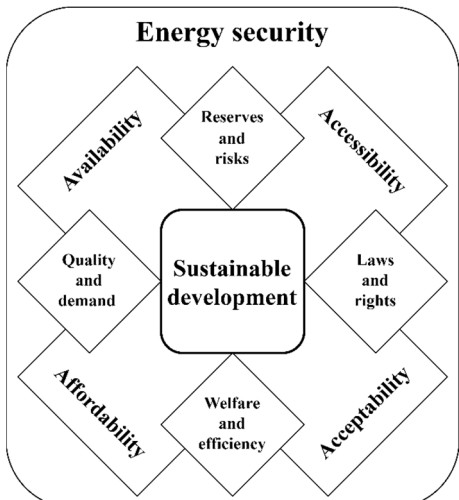

**Figure 1.** Links between energy security and sustainable development.

In recent years, energy security has become one of the critical aspects of policymaking [14]. Polona Šprajc et al. mentioned that a number of national energy agencies and international organizations have developed measurement and monitoring systems in the area of energy security and performance, e.g., the International Energy Agency (IEA) [15]. The IEA is the leading player in the dialogue on energy, providing authoritative analysis and policy recommendations to help countries provide secure and sustainable energy [16].

### 1.2. Existing Problems with Energy Supply in the European Union

The energy crisis in the European Union (EU) is an acute problem [17,18]. In the face of global instability caused by economic factors and political conflicts [19], the EU depends on imported energy sources (e.g., natural gas). Economic sanctions, world policy changes, and energy resource prices threaten the EU's energy security [20]. Introducing new technologies and transitioning to renewable energy sources (RES) became an urgent challenge to ensure the EU's sustainable and stable energy supply, reducing dependence on external supplies and overcoming the challenges of climate change and geopolitical uncertainty [21].

It is a fact that the EU is facing similar challenges, and this is not the first time [22–24]. However, based on the experience already gained, the current problems may be solved [25,26]. One possible solution is to focus on sustainable development to improve energy security and make it more flexible [27]. The EU is energy independent in part—the average energy import dependency rate stood at approximately 55% (this trend has been feeding for at least the last six years) [28,29]. In some EU countries, this figure may be well above or well below (e.g., 95% for Malta and 15% for Denmark [29]), but historically it varies according to different factors [30–32]. While the EU's member states try to diversify the energy market (e.g., diversifying gas imports away from the Russian Federation), they make themselves dependent on non-EU countries. However, improved cross-border energy network connectivity may become a solution to energy shortages [33,34]. Municipalities of different EU member states that historically depended on imported energy carriers need to move steadily towards RES and ensure energy security in parallel by taking global experience as a knowledge base [35–38].

### 1.3. Choice of Energy Supply Technologies

There are two types of approaches to heat and power or electricity generation: centralised or district [39,40] and de-centralised or individual [41,42]. The main advantages and disadvantages of centralised and individual energy generation systems are aggregated in Table 1.

**Table 1.** The advantages and disadvantages of centralised and individual systems.

| | Advantages | Disadvantages |
|---|---|---|
| Centralised systems | Ensure that the day-ahead dispatch is technically feasible and (ideally) cost-efficient [43]; One or a few large central production facilities are much cheaper than thousands of individual installations [44]; Some heat resources can only be utilized economically in district heating (e.g., excess heat from industrial production, waste heat from electricity production, heat from waste incineration, geothermal energy) [44]; District heating is the only way to supply (existing/historical) city centres (e.g., Old City of Riga, Latvia) with large fractions of RES (renovation to zero-energy houses is not possible here) [44]. | Rural electrification (in an integrated power system, rural electrification is challenging) [45]; Some buildings are overheated or underheated due to different conditions (e.g., no energy audit was carried out) [46]; There are various limitations in electric and heat networks that prevent new users from connecting [47,48]. |
| Individual systems | Independent systems with maximal flexibility; hight sustainability level [49]; Ease of decision-making in relation to development and investment; Minimal energy losses at energy transportation to consumer. | Lack of economic scale effects [50]; Relative higher emissions to produced amount of energy; Less opportunities for receiving investments. |

There are various technological solutions to increase energy security. Such solutions are applicable in both centralised and decentralised systems. Depending on the capacity scale being installed, the applications, range of resources, and degree of complexity may vary. Three classic technologies fall within the concept of sustainability in the case of both electricity and heat generation: biomass combustion units [51,52], solar energy systems [53,54], and heat pumps [55,56].

### 1.3.1. Biomass Boilers

There are several methods available to convert biomass into a useable form of energy: The foremost among them are:

Direct combustion—historically, the most common method for burning fuel in a chamber to produce heat is used in different ways depending on the technology chosen (e.g., steam-powered turbine) [57–59].

Pyrolysis—thermal decomposition of the organic matter in the absence of oxygen (this process is a relatively slow chemical reaction occurring at low temperatures to convert biomass to a more helpful fuel such as a hydrocarbon-rich gas mixture and a carbon-rich solid residue) [60,61].

Gasification—thermochemical conversion of biomass into gaseous using partial oxidation of the biomass at high temperatures [62,63].

Wood chips are the most used fuel in boiler houses that provide district heating [64]. The wood chips usually form as a residue of logging, resulting in both biodegradable wastes being used correctly. Considering the capacity of boiler houses, in many cases, it is necessary to install additional solutions to improve the system's overall energy efficiency, such as flue-gas condensers, which allow excess heat to be recovered [65–67].

### 1.3.2. Stiling Engine

For individual heating boilers, the range of efficiency-enhancing technologies is small, e.g., small-scale flue gas capacitors are not too standard a technology [68–70]. Researchers acknowledge that using another technology called the Stirling engine is possible to improve efficiency [71–74].

Stirling engine (SE) is one kind of external combustion engine which converts thermal energy into kinetic energy by heating and cooling the working gas sealed in the cylin-

ders [75]. SE has six main components: containers, piston, displacer, crankshaft, flywheel, and external heat source [76].

The SE has three configurations which are α, β, and γ [77]:

The α-type consists of two separated cylinders—the first is exposed to a hot temperature source, while the other is subjected to a cold temperature source.

The β-type consists of only one cylinder with one sealed piston and a displacer.

The γ-type is much similar to the β-type, except that the cooling chamber shifts to a separated small cylinder.

Nowadays, the number of studies on the use of SE is increasing [78].

### 1.3.3. Solar Energy Systems

Solar energy systems are based on the principle of transforming sun rays into power [79]. Solar collectors are used to generate heat, and photovoltaic (PV) panels to generate electricity [80–83].

Solar energy use in households and the commercial sector is increasing [84,85]. The change from energy consumer to energy producer and consumer (the so-called prosumer) and its impact on the way individuals use energy plays a significant role in the case of energy security [86,87].

Energy storage technologies are used to improve the efficiency of solar systems. Thermal energy storage (TES) systems can store heat or cold to be used later, at different conditions such as temperature, place, or power [88,89]. On the other hand, electrical energy storage (EES) plays three prominent roles—lowering the electricity supply costs by storing energy at off-peak rates, improving reliability at times of unexpected failures or disasters, and maintaining and improving power quality (frequency and voltage) [90].

Large-scale solar applications benefit from the effect of scale. Compared to small-scale solar systems for households or municipal buildings, the solar heat cost can be cut by at least a third. The most exciting projects for replacing fossil fuels and reducing $CO_2$ emissions are solar systems with seasonal TES in combination with biomass boilers [91,92].

Small-scale solar systems are usually installed at a faster pace. It is disturbed by several obstacles (the most important one is that in non-electrification settlements, introducing such technologies is cumbersome due to hard-to-reach energy users [93]). There are different solutions, e.g., by targeting capital constraints rather than income constraints, efficiency gains may be achieved [94]. Therefore, solar energy is usually used to cover part of consumption in individual energy systems and is often combined with other technological solutions.

### 1.3.4. Heat Pumps

A heat pump (HP) is a thermal machine that allows increasing the temperature of a heat input using a relatively low energy input [95] (in other words, HP uses electric power to generate low-temperature heat [96]). Three main types of HPs connected by ducts exist and are as follows:

Air-source—these are mainly based on fin-and-tube-type heat exchangers to extract heat from the surrounding ambient or even from the recirculating air [97] (can collect heat even under low insolation [98]).

Liquid-source—these are common in installations where internal heat sources or heat or cold reclamation is possible from the liquid (mainly water) resource [99].

Geothermal—this pumps heat to or from the ground (it uses the earth as a heat source (in the winter) or a heat sink (in the summer)) [100].

HPs are used for both heating and cooling. A HP water heater operates on an electrically driven vapour-compression cycle and pumps energy from the air in its surroundings to TES, thus raising the temperature of the water [101].

Large-scale HPs can recover and upgrade waste heat using renewable electricity, creating an essential technological lever to decarbonise thermal processes. Unfortunately, there is a lack of information and experience about meaningful applications of large-scale

HPs in practice due to missing awareness and experience from end customers, installers, and engineers. These knowledge gaps and the low level of market transparency reinforce the tendency to use traditional fossil fuel heating technologies [102].

The main obstacle for small-capacity HPs is the long payback period [103–105]. Despite the relatively high efficiency, potential users usually choose cheaper solutions. The selection of the appropriate resource depending on the potential location of the HP is an essential process.

### 1.3.5. Hybrid Systems

There are many combinations of different energy production technologies, e.g., there is a solar hybrid system that can harvest solar energy in the form of thermal energy and electricity simultaneously [106]. Such systems may also combine with liquid-source HPs [99].

Solar radiation increases the temperature of PV modules, resulting in a drop in their electrical efficiency. By properly circulating a fluid with a low inlet temperature, heat is extracted from the PV modules, keeping the electrical efficiency at satisfactory values. The extracted thermal energy can be used in several ways, increasing the total energy output of the system. Hybrid PV and thermal systems can be applied mainly in buildings to produce electricity and heat and are suitable for PV applications under high solar radiation and ambient temperature values [107].

SEs are widely used in solar energy applications due to their high power-to-weight ratio and high thermal efficiency when high temperatures are present due to solar concentration. However, the efficiency of SDSE models faces many challenges, such as attaining high regenerator efficiency, high solar radiation absorption at the receiver, and low temperatures at the rejector side [108].

The PV boiler application is reliable due to minimum additional cost and low LCOE values. However, the hybrid PV-SE-boiler system has the lowest LCOE and payback period. With the use of μ-cogeneration systems in heating, the electricity requirement of the auxiliary equipment in the heating system implements the SE and PV as a solution for individual heating systems, especially in areas away from the grid (e.g., rural areas) [109].

### *1.4. Research Gap*

An advanced literature analysis of similar studies found that a larger proportion of studies include general solutions for introducing energy security in a conditional location. Some studies offer to solve problems on the political side in the sector, but there are few studies that look in depth at the technical aspects of technology. In case studies, most solutions can be found in the context of centralised energy supply. The current study pertains to introducing energy security in the context of individual energy supply, which includes increasing energy independence not only in regional situations but also in local energy supply situations.

### *1.5. Main Objective*

This study aimed to evaluate possibilities to ensure 100% self-sufficiency of thermal energy and electricity consumption using the Stirling engine to increase energy security for individual energy consumers (for information about the selected case study, see Section 3). This study aimed to answer the question of whether there is a possibility of transforming the energy supply process by promoting increased energy security and independence while not reducing energy demand, as well as creating high economic and environmental indicators.

The research focuses on developing alternative scenarios for decision-making (see Section 3.4). Scenarios are analysed using the complex method developed (see Section 2.1). The method proposed in this study to promote energy security is developed based on a case study.

### *1.6. Hypothesis*

Using economic, environmental, and technical indicators, the potential of the Stirling engine for integration into the conditional energy system can be better assessed than if only technical aspects are taken into account for the analysis.

## 2. Materials and Methods

### *2.1. Research Method*

The complex method is designed to assess scenarios for increasing energy security. The method is based on a variety of quantitative and qualitative research methods suitable for research in the context of energy security [110]. The method is universal and can be used for the analysis of different energy users (see Figure 2). The first step is to assess the current situation. Data availability (user data or from energy audit) plays an important role in creating the right preconception and further creating appropriate scenarios for development. The analysis of thermal energy and electricity consumption provides an answer regarding possible solutions for increasing energy security. Before creating an alternative scenario with any available technology, it is primarily necessary to assess the feasibility of reducing existing thermal energy and electricity consumption. Reducing energy consumption is a more important precondition for increasing energy security. Once required capacities have been defined, it is possible to select appropriate energy technologies for operating times and other technical generators. If more suitable technologies exist, more scenarios can be developed. If, after an assessment of the current situation, it was found that the alternative scenario is not necessary or that transition to new technologies is not possible, then the proposed solution may contain recommendations to improve the existing technology.

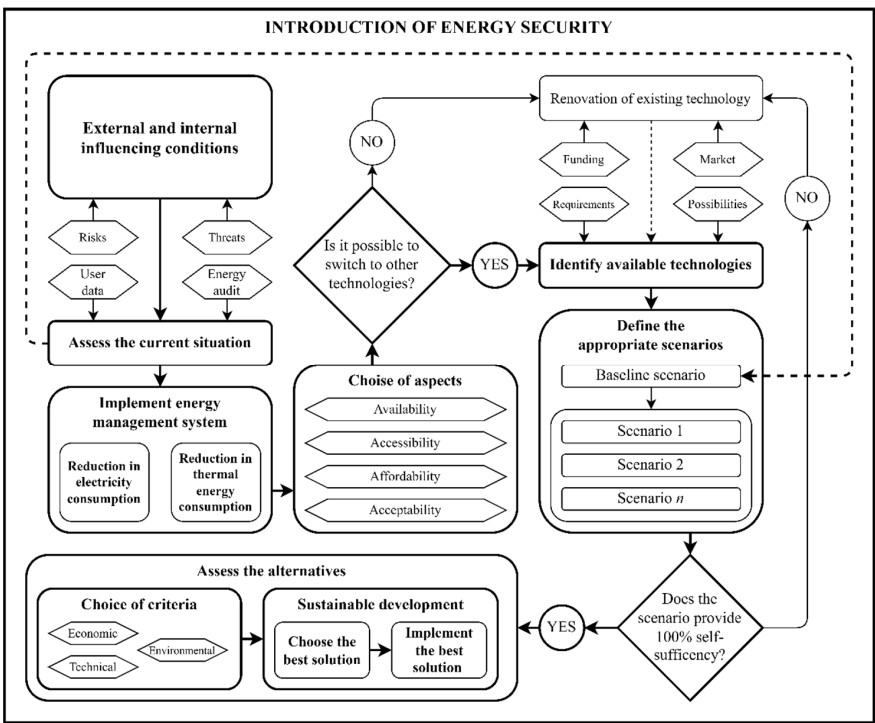

**Figure 2.** Energy security evaluation scheme.

All defined scenarios are compared with each other, as well as with the base scenario, which describes the current situation. The user can choose the criteria by which the analysed scenarios will be compared. It is possible to define the number of criteria and limit values individually for each user. The mandatory criterion is the possibility of providing 100% self-sufficiency with a chosen scenario and technological solution. If this criterion is met, then further analysis of the scenario using other criteria may follow. Criteria can be from different dimensions—technological (the output power, energy consumption, etc.),

economical (cost of energy production, the required amount of investment, payback period, etc.), environmental ($CO_2$ emissions, air polluting emissions, generated waste, etc.). Criteria can have equal weight but different importance or influence.

### 2.2. Parameters and Values Used in This Study for Calculation

Technical-economic variables and dependencies were selected for the calculation of scenarios. Variables are based on data from the literature, the situation in the energy market, and expert assumptions. Variable values are shown in Table 2.

**Table 2.** Assumed values of the technical-economic parameters included in this study.

| Variable | Value | Unit |
|---|---|---|
| Specific cost of Stirling engine | 750 | EUR/kW |
| Specific cost of solar PV | $25,883x^{0.374}$ | EUR/kW |
| Specific cost of heat pump | 296 | EUR/kW |
| Specific cost of solar collector | 295 | EUR/m$^2$ |
| Specific cost of heat accumulation system | 216 | EUR/m$^3$ |
| Efficiency solar PV | 21.1 | % |
| Solar PV loss factor | 95 | % |
| Solar collector efficiency | 77.5 | % |
| Loss factor of the solar collector system | 90 | % |
| Specific cost of natural gas | 150 | EUR/MWh |
| Additional cost for energy production from natural gas | 20.0 | % |
| Specific cost of wood pellets | 104 | EUR/MWh |
| Additional cost for energy production from wood pellets | 40.0 | % |
| Electricity tariff | 350 | EUR/MWh |
| Specific cost of electricity (sell) | 280 | EUR/MWh |

x—technology capacity, kw.

The efficiency of energy production using SE and natural gas boilers is dependent on the variables of energy technology load and is described in Section 3.4.3. Considering the feasibility study carried out, the authors additionally introduced two factors that determine the unpredicted additional costs: for natural gas technologies, it amounts to 20%, and for SE, it amounts to 40%.

The emission amount from biomass and natural gas combustion, as well as electricity production, was calculated based on emission factors. The emission factors are shown in Figure 3 and are based on data from the literature [111–113].

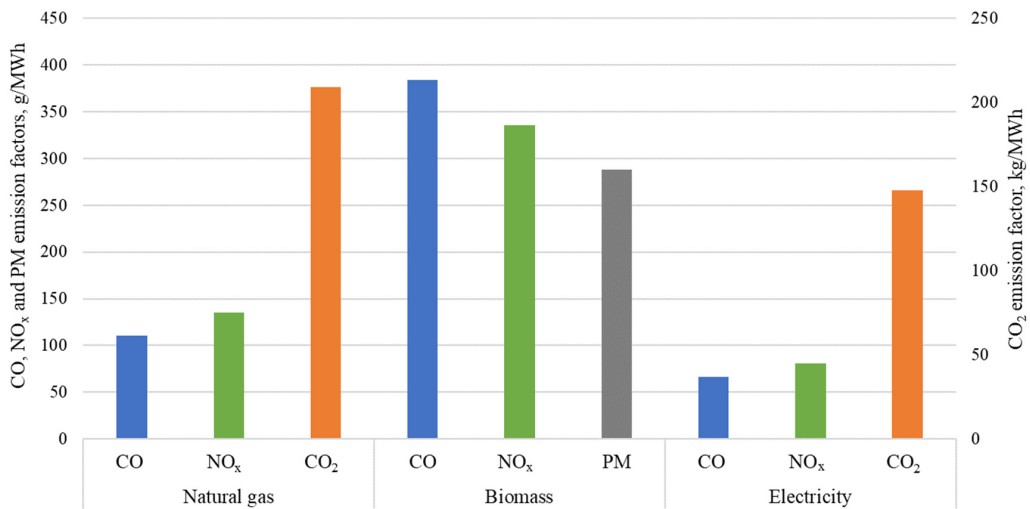

**Figure 3.** Assumed emission factors for energy sources included in this study.

All the variables introduced were used to calculate four criteria—total investments, specific energy cost, payback period, and emission amount. The criteria were calculated for all scenarios described for each municipal building. The calculations under the scenarios were made for two cases, where the actual heat load is equal to 100% of the expected load and 80% of the expected load.

### 2.3. Equations Used

The indicator of costs during the energy production process is called specific energy cost. For the baseline scenario (*BS*), this indicator is calculated based on data from Table 2 and using the following equation:

$$SEC_0 = \frac{TEC}{E + Q} = \frac{(E{\cdot}ET) + \left(\frac{Q}{\eta_G}{\cdot}SC_G\right)}{E + Q} \tag{1}$$

where

| | |
|---|---|
| $SEC_0$ | Specific Energy Cost for Baseline Scenario (Scenario 0), EUR/MWh |
| $TEC$ | Total Energy Cost, EUR |
| $E$ | Annual Electricity Consumption, MWh |
| $Q$ | Annual Thermal Energy Consumption, MWh |
| $ET$ | Electricity Tariff, EUR/MWh |
| $\eta_G$ | Efficiency of the Natural Gas Boiler, % |
| $SC_G$ | Specific Cost of Natural Gas, EUR/MWh |

For alternative scenarios (*S1*, *S2*, *S3*), this indicator was calculated using the following equation:

$$SEC = \frac{TEC}{E + Q} = \frac{(B_F{\cdot}SC_F - EP_{EX}{\cdot}SC_E)}{E + Q} \tag{2}$$

where

| | |
|---|---|
| $SEC$ | Specific Energy Cost for Alternative Scenarios, EUR/MWh |
| $B_F$ | Fuel Consumption in Energy Equivalence, MWh |
| $SC_F$ | Specific Cost of Fuel, EUR/MWh |
| $EP_{EX}$ | Excess Electricity Produced, MWh |
| $SC_E$ | Specific Cost of Electricity (sell), EUR/MWh |

Total investments are an indicator of the amount of investment needed to install a technology and provide the required capacity (electric or thermal). Depending on the quantity of technologies expected, the indicator is calculated using the following equation:

$$TI = \sum(C{\cdot}SC_T) \tag{3}$$

where

| | |
|---|---|
| $TI$ | Total Investments, EUR |
| $C$ | Capacity Required, kW |
| $SC_T$ | Specific Cost of Technology, EUR/kW |

The payback period is an indicator of time is needed to project will remain profitable. Based on values calculated by Equations (1)–(3), the indicator is calculated using the following equation:

$$PP = \frac{TI}{SEC_0 - SEC} \tag{4}$$

where

| | |
|---|---|
| $PP$ | Payback Period, year |

The amount of emissions produced was calculated as the environmental impact in each scenario. In the case of emissions from the production of $Q$, the calculation was made using Equation (5) and, in the case of generation of $E$, Equation (6).

$$M_{EM} = B_F \cdot \frac{EF}{1000} \tag{5}$$

Here,

$M_{EM}$     Annual Amount of Emissions, g (for $CO_2$–kg)
$EF$       Emission Factor (see Figure 3), g/MWh

$$M_{EM} = E \cdot \frac{EF}{1000} \tag{6}$$

## 3. Analysis of Input Data for Defining Scenarios for Case Study

### 3.1. General Description of the Existing Situation

The town of Ādaži is the administrative center of Ādaži Municipality. Only local municipal buildings were considered as part of this study. The town historically uses natural gas for district heating only in households. Hot water in households and municipal buildings is prepared individually with electric and natural gas (sometimes–diesel) boilers.

The buildings included in this study are marked in Figure 4. Table 3, on the other hand, presents their main parameters. The table contains available information on electricity ($E$) consumption (including specific annual $E$ consumption), thermal energy ($Q$) consumption (including specific annual $Q$ consumption), the renovation state and available roof space.

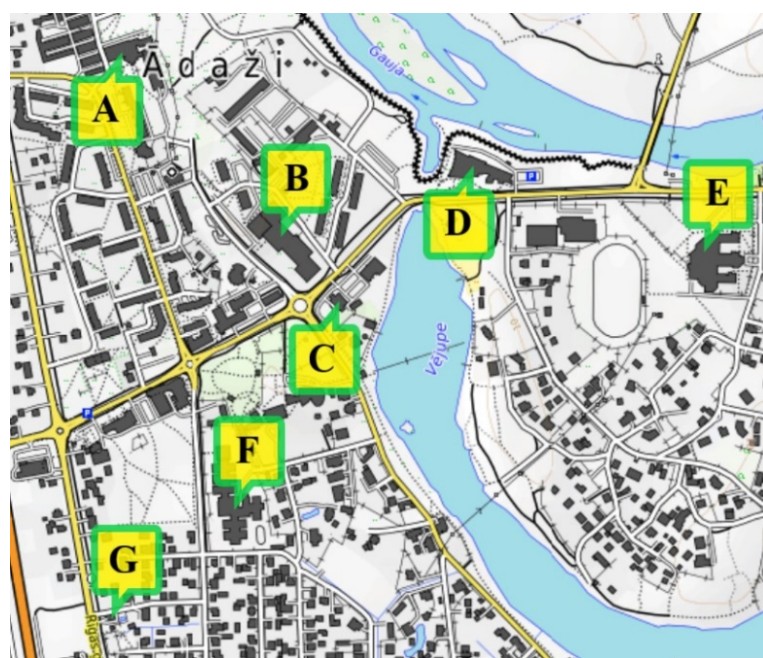

**Figure 4.** Boundaries of the town of Ādaži included in this study (location of selected local municipal buildings).

Apart from the availability of district heating in the town, municipal buildings were historically not connected to the heat network. There are two main types of problems: distance and heat load. When studying the current situation, the authors of the work have found that buildings located relatively close to the heat network have a relatively high heat load, which cannot be provided with existing pipes because the internal diameter is not large enough. On the other hand, buildings which are relatively far from the heat network cannot be connected because the construction of the new pipeline is a high-cost scenario,

given that the theoretical calculated losses in the new pipeline will be too high, making the payback period longer.

**Table 3.** Technical parameters of selected municipal buildings.

| Municipal Building | Sign | Heating Area, m$^2$ | E (Average), MWh/Month | Q (Average), MWh/Month | Specific Annual E, kWh/m$^2$ | Specific Annual Q, kWh/m$^2$ | Renovation State | Available Roof Area, m$^2$ |
|---|---|---|---|---|---|---|---|---|
| Office building | A | 333 | 3.8 | 15.6 * 1.1 ** | 131 | 328 | Partly renovated | 423 |
| Primary school | B | 8724 | 17.0 | 54.1 * 1.5 ** | 23 | 51 | New | 472 |
| Council building | C | 729 | 1.9 | 12.8 * 2.3 ** | 32 | 163 | Partly renovated | 81 |
| Cultural Centre | D | 6285 | 19.3 | 113.0 * 12.8 ** | 37 | 164 | New | 910 |
| Secondary school | E | 16,186 | 38.0 | 267.2 * 44.4 ** | 28 | 140 | Partly renovated | 1761 |
| Kinder-garten | F | 4138 | 8.6 | 109.6 * 21.6 ** | 25 | 238 | Renovated | 480 |
| Police department | G | 194 | 1.1 | 7.8 * 1.4 ** | 66 | 323 | Partly renovated | 57 |

*—heating season, **—off-heating season.

### 3.2. Thermal Energy (Q) Consumption

The *Q* consumption of selected buildings depends mainly on the outdoor air temperature (see Figure 5). The correlation between *Q* consumption and outdoor air temperature was determined based on available historical data. In general, a relatively good correlation can be observed for all buildings with an average coefficient of determination $R^2$ of 0.71. However, for almost all the buildings, there are individual cases when the *Q* consumption is higher than the established correlation curve. A typical reason for such dispersion is an incorrectly adjusted heating system that doesn't ensure the correct amount of supplied *Q*. As a result, the building receives excess *Q*, increases indoor temperature and heat losses, and misuses energy resources.

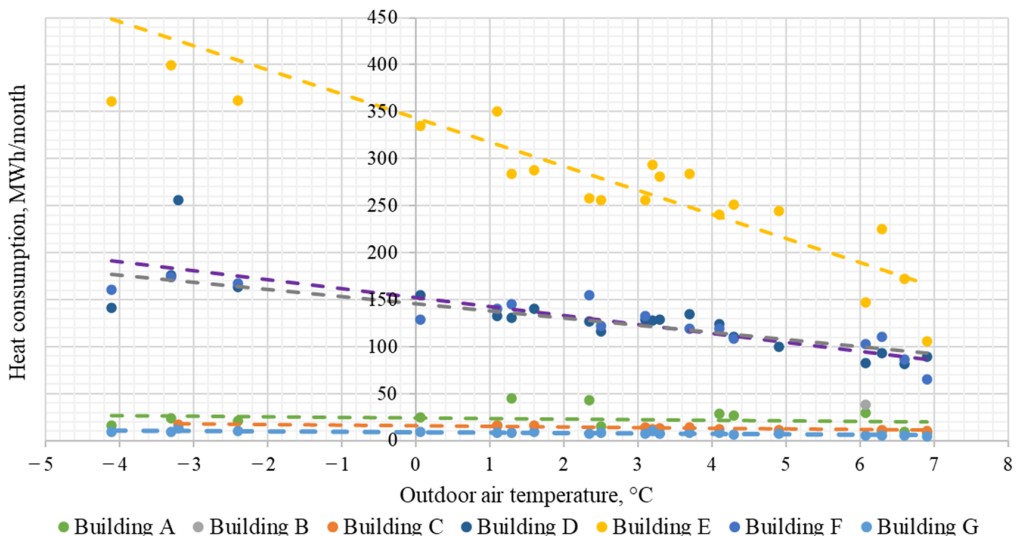

**Figure 5.** Correlation of thermal energy consumption and outdoor air temperature of selected municipal buildings during the heating season.

### 3.3. Electricity (E) Consumption

The *E* consumption of selected buildings is significantly lower than the demand for thermal energy, but its significance is equally important in the context of a common energy supply. The *E* consumption profile of buildings differs significantly, mainly due to the needs and characteristics of different buildings (school, educational; office building, commercial, etc.; see Figure 6). Electricity is primarily used for lighting purposes. Considering the location of the selected area, the seasons are changing in the region. As seasons change, the availability of natural light during the day also varies—availability is more extended in the summer months compared to the winter months. However, not all buildings depend on the availability of natural light, so there is no such marked change in *E* consumption during the year. Some buildings (E and F) have experienced a marked increase in *E* consumption during the winter months. Building D, on the other hand, has an *E* consumption peak in summer. There are also buildings for which peaks are not explained properly: e.g., Building B has *E* consumption peak between September and December. Buildings C and G have minimum fluctuations in *E* consumption, which is a good precondition for a transformation to efficient energy system.

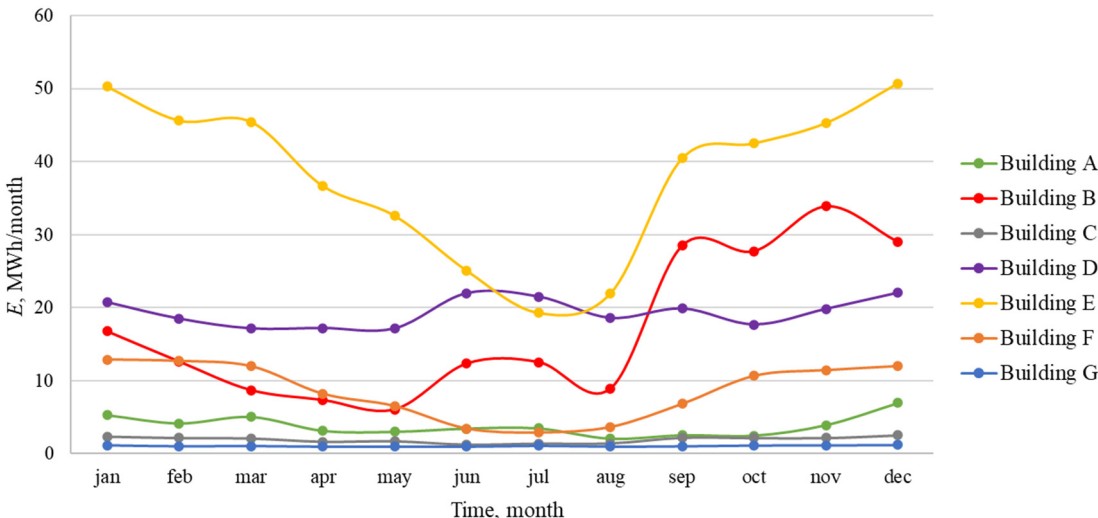

**Figure 6.** Annual electricity consumption of selected municipal buildings.

### 3.4. Development Scenarios

Based on the existing situation analyses in the town of Ādaži and technological examples found in the literature, different scenarios were developed to improve energy security in the town of Ādaži. Four scenarios with different selected technologies as well as a scenario with reduced *Q* and *E* load were analyzed together. Each scenario envisages 100% self-sufficiency of *Q* and *E* consumption in municipal buildings.

#### 3.4.1. Baseline Scenario (BS)

The current situation within the scope of this study is a baseline scenario comparing with alternatives development scenarios. The baseline scenario includes 100% use of natural gas in all buildings for *Q* production and *E* purchasing from the grid.

#### 3.4.2. Biomass + SE (S1)

The first alternative scenario is based on biomass usage for *Q* and *E* production. Biomass is a widely available energy source in Latvia since more than half of the country's territory is covered by forests. There are many technologies available based on biomass usage. A boiler with SE was selected as biomass technology considering the characteristics of the selected case study with the necessity of production of both Q and E, relatively small capacities, as well as performed literature review.

One of the main criteria for the SE is the ratio between the amount of *Q* and *E* produced. This ratio may vary significantly depending on the constructive solution. This ratio can range from 1.7 to 8.8 [114]. Like the other biomass-based technologies, the efficiency of the SE depends on the load. The maximum efficiency is achieved at a nominal load (100%) and is equal to 87.7%. As the load in efficiency decreases significantly, reaching 50.5% at ¼ of the nominal load. Natural gas boilers also tend to have this trend, but there is less efficiency reduction (see Figure 7) [114,115]. This aspect demonstrates that to achieve rational fuel consumption, it is important to ensure that the consumer has as smooth a load as possible and less time when energy technology operates at reduced loads. The calculations considered variable efficiency for both the SE and the natural gas boiler dependency on load. Regression equations were used, which were obtained and are shown in Figure 7. The results obtained are much more reliable compared to the data presented in other studies.

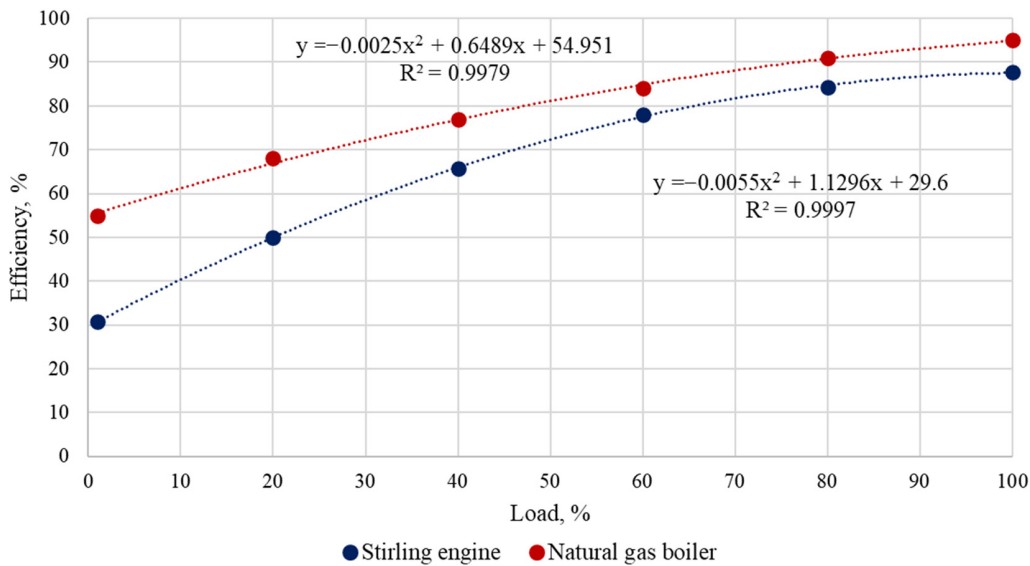

**Figure 7.** Energy technology efficiency dependency on load.

### 3.4.3. Solar PV + HP (S2)

The second scenario is based on solar energy usage for energy production. Solar PV technologies were selected for *E* production and to cover building consumption. Considering the characteristics of Latvia's climate and the significant difference in solar radiation intensity between the summer and winter months, the amount of *E* produced also differs significantly between months. The possible amount of *E* produced significantly exceeds the self-consumption of buildings in several months. Therefore, in the scenario, as an additional technology, a HP is used for each building to convert excess *E* into *Q*. Two factors are important for the installation of PV panels on building a roof: a south-facing roof and the available roof area. The authors have determined the available roof area (see Table 3) considering the roof angle that was used for further calculations. The authors have investigated the area covered by this study's boundaries, and the characteristics of each building's technical equipment and found that air-source is the most appropriate HP technology in all cases.

By installing the maximum amount of available solar PV considering the roof area of the buildings, a surplus of electricity is generated in some summer months, even after using a certain amount of electricity for the HP's needs for heat energy production. This is especially true for buildings that have low *Q* consumption in the summer months. Consequently, a surplus of produced *E* is formed. In the framework of this study, it is considered that the surplus of this *E* is transferred to the common network, and the corresponding cash income is received from the *E* sold. To cover the remaining demand for *Q* and *E*, a SE with suitable capacity is selected.

### 3.4.4. Solar Collectors + TES (S3)

The third scenario is based on solar collectors using TES for heat demand covering. Additionally, *Q* and *E* are produced by SE. The necessary area of solar collectors was calculated to cover the average heat demand at summer mounts. It is not justified to install the maximum volume of collectors to create significant excess *Q*. In cases where the roof area is available after the installation of the necessary mount of solar collectors, the amount of possible solar PV installation is additionally calculated. To cover the remaining demand for *Q* and *E*, a SE with suitable capacity is selected.

## 4. Analysis of Output Data and Scenario's Simulation Results

### 4.1. Reduction of Thermal Energy and Electricity Consumption

#### 4.1.1. Thermal Energy (Q) Consumption

Based on the analysis of *Q* consumption (see Section 3.2), options for reducing it were identified for the selected buildings. Figure 8 shows that an established regression curve (red curve) was selected for the benchmark, which describes the *Q* consumption depending on the outdoor air temperature in the current situation. Accordingly, all consumption above the correlation curve (red dots) is considered unreasonable and reduced to the benchmark (green dots). The *Q* consumption below the curve is not changed (blue dots). This results in a new regression curve (blue curve) that describes the *Q* consumption depending on outdoor air temperature after the benchmark has been introduced. Figure 8 represents the thermal energy consumption of Building B. The calculation approach described was also applied to other buildings.

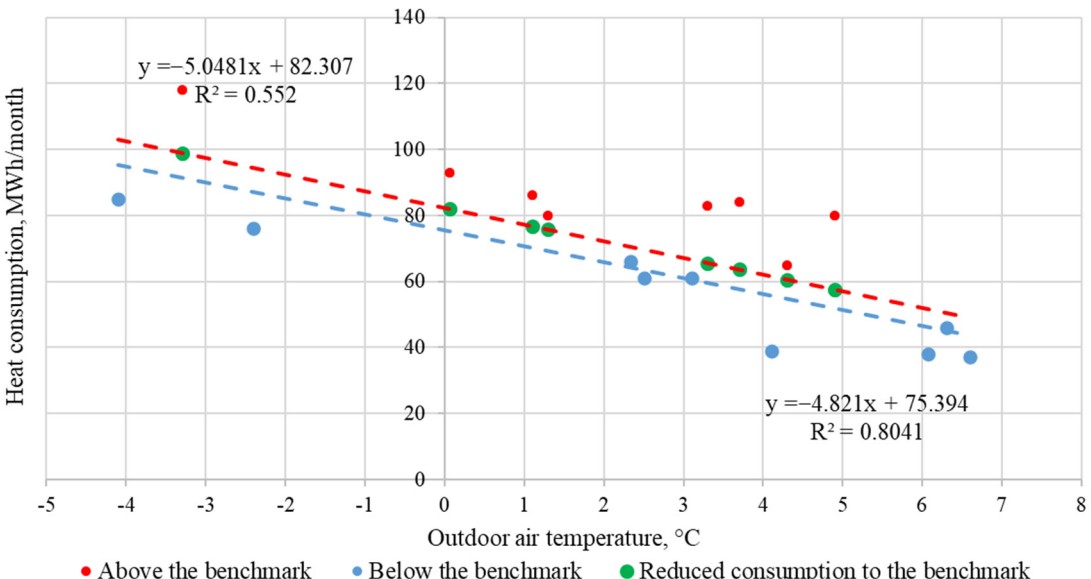

**Figure 8.** Modified correlation of thermal energy consumption and outdoor air temperature of selected municipal buildings during the heating season.

Considering the correlation between *Q* consumption and outdoor air temperature, a heat load diagram has been created (see Figure 9). The heat load diagram shows the difference between the *Q* demand of different buildings, but the common trend is that the heat load peaks are relatively short in comparison to all heating seasons. At the same time, the influence of these peaks on the heating system is significant. The capacity of energy technologies is typically chosen to fully cover the peak hours of short-term *Q* consumption. As a result, energy technologies work with reduced heat load during most of the heating season. In addition, the power selection of energy technologies according to the peak heat load contributes to increased capital costs during installation. Some buildings have *Q* consumption in the summer period for hot water preparation. The heat load in the summer period is very low in relation to the peak heat load in the winter period. Such differences in

heat load create an additional challenge for the creation of a sustainable heating system. Therefore, the reduction of $Q$ consumption has significant benefits for the creation of an efficient energy supply system.

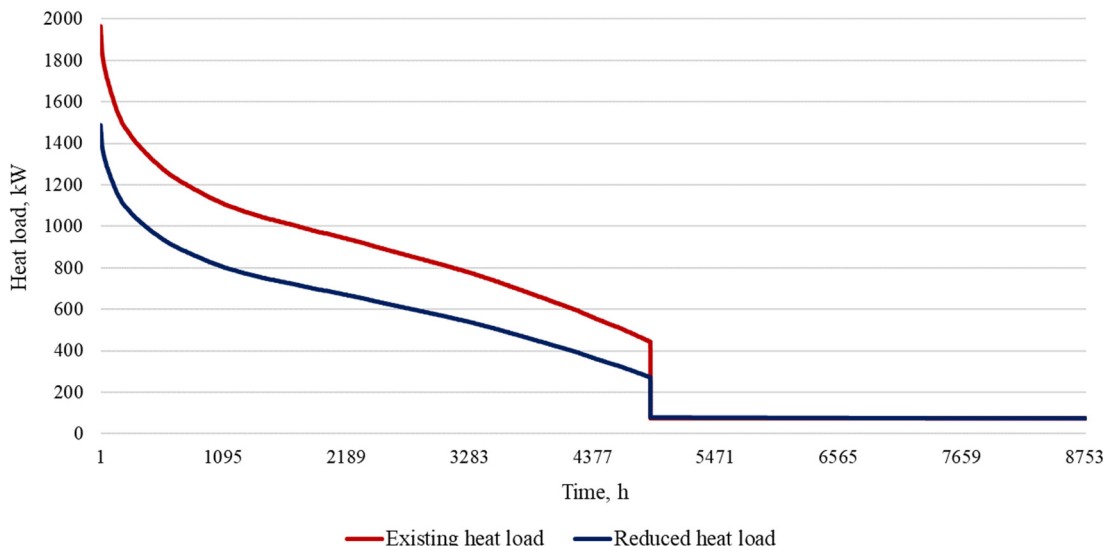

**Figure 9.** Annual total heat load of selected municipal buildings.

### 4.1.2. Electricity (E) Consumption

In the case of $E$, a benchmark for reducing $E$ consumption is also applied. Monthly $E$ consumption was used as a benchmark for each building, which may not be higher than the average annual $E$ consumption. This limits maximum $E$ consumption and peak electrical loads. There is also a decrease in average annual $E$ consumption (see Figure 10). The biggest drop in $E$ consumption is seen in the winter months. The maximum reduction is in December, which amounts to 34.7 MWh. At the same time, in different months (April, May, August), total $E$ consumption remains unchanged.

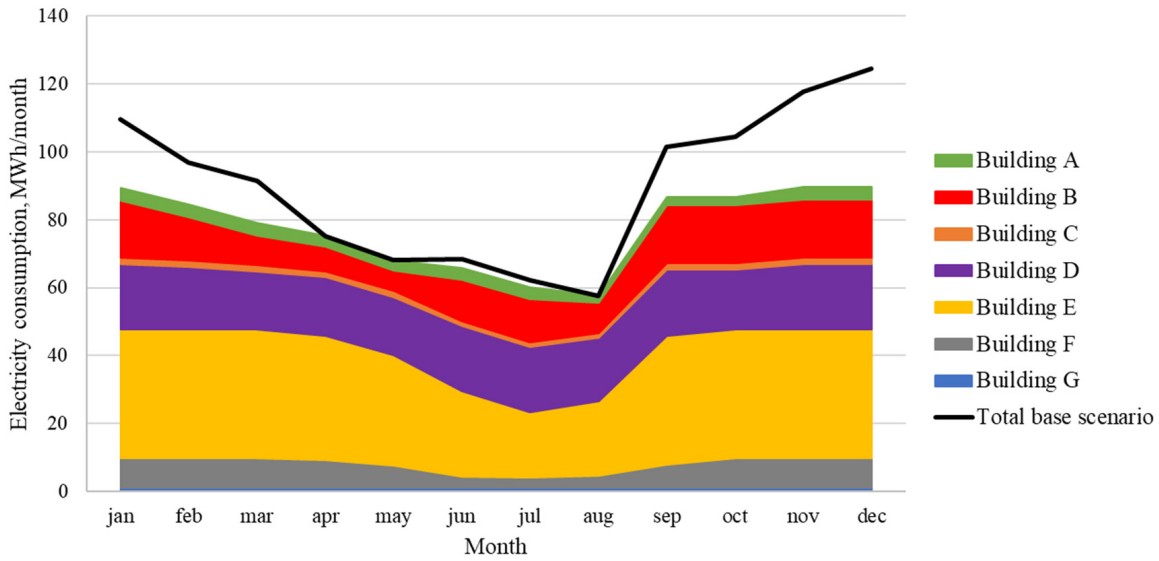

**Figure 10.** Annual electricity consumption of selected municipal buildings after introduction of energy efficiency measures compared to the current situation.

### 4.1.3. Energy Reduction Possibilities

Applying established benchmarks reduces both total $Q$ and $E$ consumption and maximum heat and electrical loads. Figure 11 shows the relative decrease between the current

situation and the benchmarks. The decrease was observed to vary significantly between buildings. This means that there are consumers for whom heat or electrical loads are already balanced. The largest reduction can be achieved in *Q* consumption (by 26.0% in total) and the lowest reduction in *E* consumption (by 13.4% in total). The reduction in peak loads is significantly higher both in terms of heat load (31.3% on average) and in terms of electrical load (28.4% on average). This is an important aspect both in the context of increasing energy security and in establishing an efficient energy system, as it reduces the necessary installation capacity of energy technologies. In addition, the efficiency of energy technologies increases (see Figure 7).

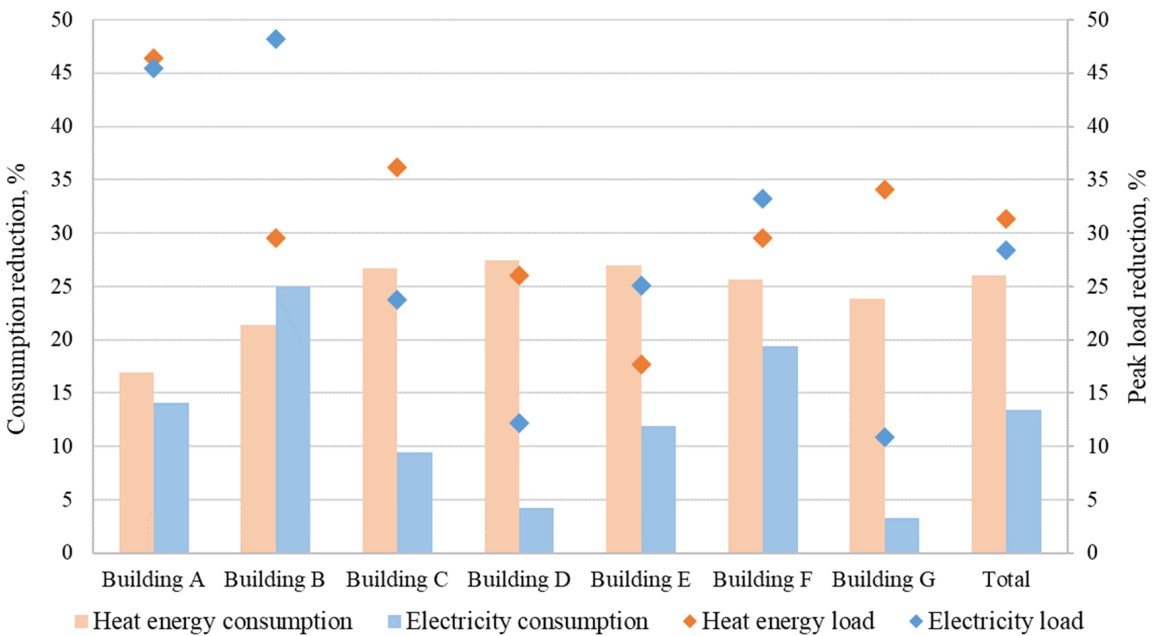

**Figure 11.** Thermal energy and electricity consumption and maximal load reduction possibilities.

*4.2. Assessment of Scenarios*

The data obtained on the consumption of thermal energy and electricity, as well as maximum loads, were further used to assess energy supply scenarios. Scenarios differ on two levels:

- by load (baseline or existing load and reduced load (based on established benchmarks);
- by energy technology used (natural gas, Stirling engine, solar PV + heat pump, solar collectors + thermal energy storage).

The description of the scenarios and their characteristics are shown in Table 4.

**Table 4.** Characteristics of selected scenarios.

| Energy Consumption | Scenario Name | Energy Technologies Used |
|---|---|---|
| Baseline energy consumption | Base load, BS | Natural gas boiler, electricity from the grid |
| | Base load, S1 | Stirling engine |
| | Base load, S2 | Solar PV + Heat pump. Stirling engine to cover residual load |
| | Base load, S3 | Solar collectors + storage. Stirling engine to cover residual load |
| Reduced energy consumption | Reduced load, BS | Natural gas boiler, electricity from the grid |
| | Reduced load, S1 | Stirling engine |
| | Reduced load, S2 | Solar PV + Heat pump. Stirling engine to cover residual load |
| | Reduced load, S3 | Solar collectors + storage. Stirling engine to cover residual load |

*4.3. Specific Energy Cost*

Specific energy cost was calculated for each scenario included in this study. The results show that in each alternative scenario (*S1*, *S2,* and *S3*) this indicator is lower than in the current situation (Scenario *BS*). In Figure 12, the dispersion of specific energy costs is reflected between all selected buildings considered in all scenarios. Scenario *S1* with the SE has the highest dispersion, suggesting that this technology can be suitable and reach high results under certain preconditions (smoothed *Q* and *E* consumption without peaks, summer heat load) and appropriate *Q* and *E* consumption ratios. Scenario *BS* is the most disadvantageous option (the median indicator is 241 EUR/MWh at the base load and 245 EUR/MWh at the reduced load). The most advantageous alternative is Scenario *S2*: By introducing the technologies proposed in this scenario, the specific energy cost will decrease on average to 101 EUR/MWh at the base load and 108 EUR/MWh at the reduced load. The reduction in load makes it possible to reduce energy consumption, but this leads to an increase in specific costs. The increase in the scenario considered is not significant—it amounts to, on average, between 4 and 10 EUR/MWh.

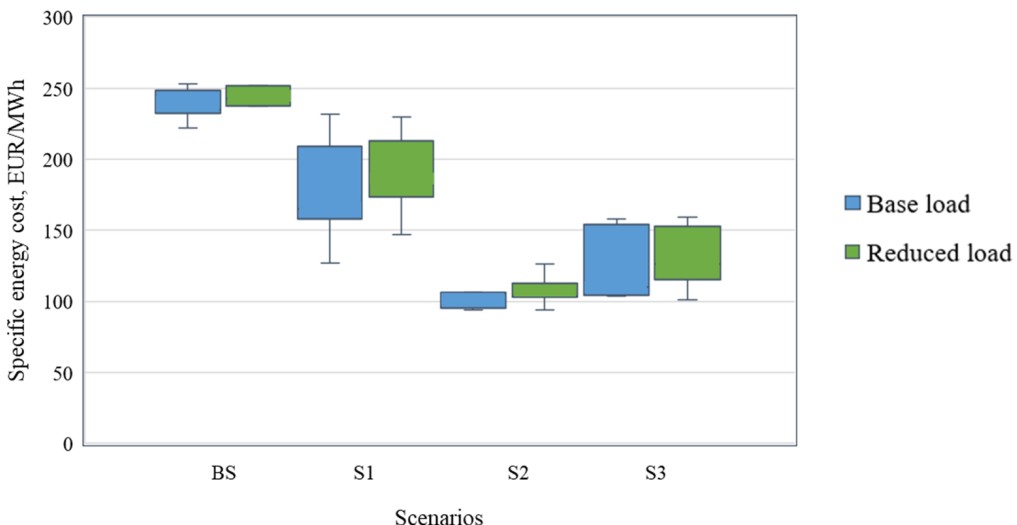

**Figure 12.** Specific energy cost of selected scenarios.

*4.4. Total Investments*

Scenarios *S1*, *S2*, and *S3* require new energy technologies that require investments. Investments were calculated based on the data presented in Table 3. Scenario *BS* is based on the current situation and energy technologies used, so no additional investments are necessary. Figure 13 shows cumulative investments—the sum of the values for each selected building in each scenario. The higher the *Q* and *E* consumption of the building, the higher the investments. The lowest indicator is for Scenario *S1*, as it expects to use only one hybrid technology—a pellet boiler with a SE (945 TEUR at the base load and 674 TEUR at the reduced load). The indicator represents the cost of both the installation of the pellet boiler and the Stirling engine. There are significantly more investments in Scenarios *S2* and *S3*, as in addition solar technologies are used—solar PV with HP in Scenario *S2* and solar collectors with TES in Scenario *S3*. The highest cumulative investments are predicted for Scenario *S3* (2110 and 1856 TEUR at base and the reduced load, respectively). Scenario *S2*, the most cost-effective from an operational point of view, is twice as expensive as Scenario *S1*. In scenarios with a reduced load, capital formation is lower than in scenarios with base load.

*4.5. Payback Period*

The payback period is an indicator that helps characterize the viability and profitability of the scenario—the shorter it is, the sooner the realization of the scenario will cover the investments. The payback period depends on the amount of the investments and the cost of

producing energy. The method for calculating the simple payback time was taken as a basis. The lower the investment and production costs, the shorter the payback period. Scenario *BS* does not involve additional investments, so there is also no time for investments to be paid back. In Figure 14, given the amount of the investments, on average, the shortest payback period, considering all municipal buildings, is for Scenario *S1*, which is equal to 3 and 4 calendar years at the base and reduced loads, respectively. This is due to lower investments, as the cost of producing energy for Scenarios *S1* is higher than for Scenarios *S2* and *S3*.

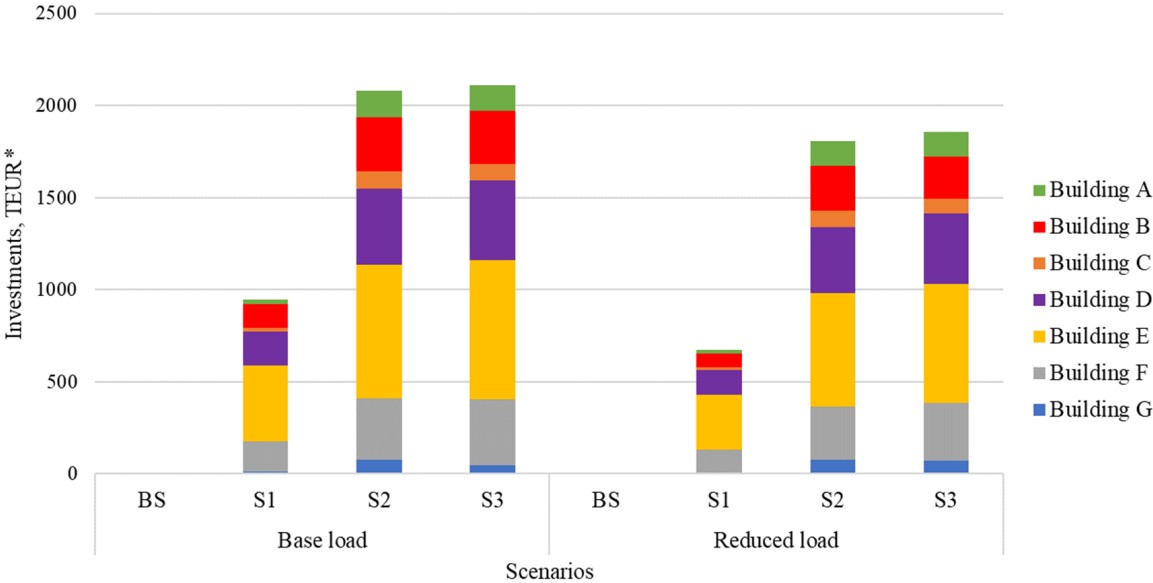

**Figure 13.** Total investment for scenario implementation (* TEUR—Euros in units of thousands).

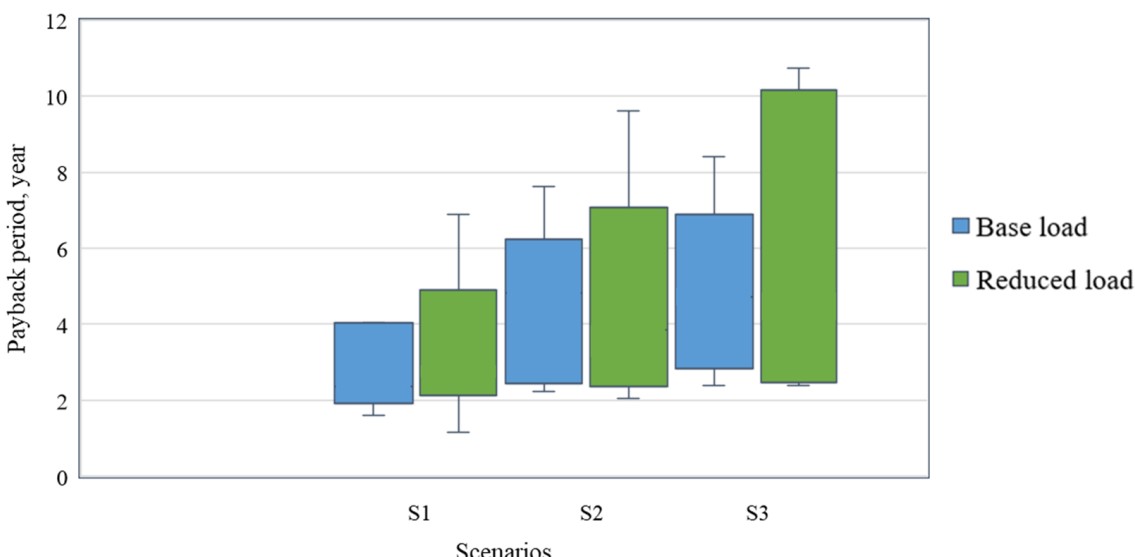

**Figure 14.** Payback period of selected scenarios.

For different buildings, the introduction of Scenario *S1* will pay back in 9 years. Buildings with the longest payback period are those for which the ratio of thermal energy to electrical energy consumption is significantly different from the ratio of heat to electrical load of the SE; energy consumption is expressed in peaks or no summer thermal energy consumption. The longest payback period is for Scenario *S3*, with average values at the base and reduced loads accounting for 5 and 6 calendar years, respectively. For Scenario *S2*, the indicator is between 4 and 5 calendar years at the base and reduced loads, respectively.

In scenarios with a reduced load, the payback period is slightly higher than in scenarios with a base load. This is due to slightly higher specific energy costs.

In the following research, an in-depth analysis of the financial key performance indicators (KPIs) should be included to gain insight into the energy scenarios and perceive the economic sustainability of the investment.

*4.6. Enviermental Impact*

4.6.1. Emission Factors

As mentioned above, this study assessed each scenario not only based on technical preconditions and economic factors but also analysed the environmental impact. The use of different fuels produces different environmental impacts. Biomass for Scenarios *S1*, *S2*, and *S3* produces CO, $NO_x$, and PM emissions since it was assumed that the Stirling engine was installed in a pellet boiler. PM emissions were formed from wood pellet combustion. These emission factors can be seen in Table 5. When comparing emissions from biomass and natural gas, it can be concluded that the burning of natural gas produces less CO and $NO_x$ emissions per energy unit. However, the amount of $CO_2$ is significant. On the other hand, biomass is considered to be a carbon-neutral fuel. It is important to note that the emission factors set out in the table refer to the total amount of energy produced, which includes both *Q* and *E* in a certain proportion. Given that, in the case of biomass, this energy resource is used at the same time, it was decided not to separate emission factors for each type of energy but to attribute it to the total amount of energy.

**Table 5.** Emission factors depending on scenarios.

| Scenarios | | CO, g/MWh | $NO_x$, g/MWh | PM, g/MWh | $CO_2$, kg/MWh |
|---|---|---|---|---|---|
| Base load | BS | 117 | 142 | 0 | 224 |
| | S1 | 521 | 456 | 391 | 0 |
| | S2 | 378 | 331 | 283 | 0 |
| | S3 | 416 | 364 | 312 | 0 |
| Reduced load | BS | 119 | 144 | 0 | 229 |
| | S1 | 536 | 469 | 402 | 0 |
| | S2 | 362 | 317 | 271 | 0 |
| | S3 | 434 | 380 | 326 | 0 |

4.6.2. Total Amount of Emissions

The overall environmental impact of the scenarios depends on the type and amount of energy resource used. The emission factor values given in Table 5 were used to calculate the total amount of emissions, knowing the total energy consumption for each scenario. Scenarios with a reduced load have lower energy consumption, so the total amount of emissions produced is also lower. In Figure 15, the use of natural gas in Scenario *BS* results in the quantity of CO and $NO_x$ being 691 and 534 kg per year at the base load and 841 and 649 kg at the reduced load. Compared to biomass (Scenario *S1*), natural gas CO emissions are about 4 times less, while $NO_x$ emissions are about 3 times less. In addition, biomass also produces a significant amount of PM emissions. It should be considered that high amounts of $CO_2$ are released during the burning of natural gas. At the base load in Scenario *BS*, 1326 tons of $CO_2$ emissions are produced, while 1026 tons of $CO_2$ emissions are produced at the reduced load. In Scenarios *S2* and *S3*, the use of solar technologies reduces biomass consumption, which also reduces air pollution. The lowest total amount of CO, $NO_x$, and PM is produced in Scenario *S2* among alternatives: 2222 kg of CO emissions, 1944 kg of $NO_x$ emissions, and 1667 kg of PM emissions at the base load and 1644 kg of CO emissions, 1438 kg of $NO_x$ emissions, and 1233 kg of PM emissions at the reduced load.

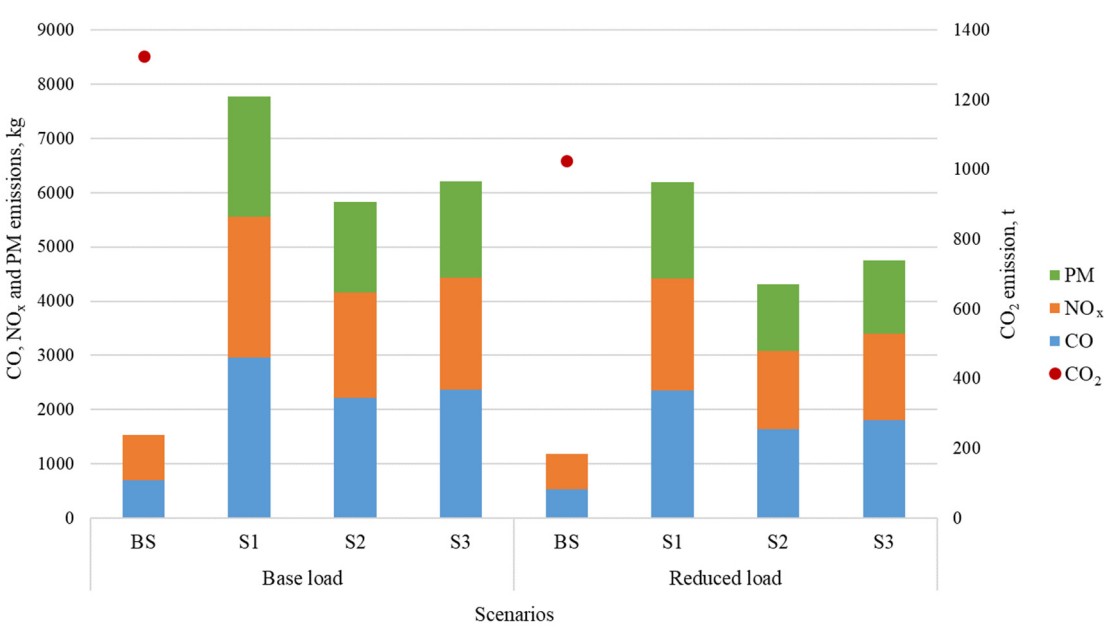

**Figure 15.** Total amount of emissions of selected scenarios.

## 5. Discussion

The aim of this study was to evaluate the possibilities of increasing energy security in municipal buildings in the event of an energy crisis. For this purpose, a complex method was developed, which includes several interrelated steps: (1) evaluation of the current situation; (2) reduction of current energy consumption; (3) selection of energy technologies and definition of alternative scenarios; (4) assessment of selected scenarios. The method was a braised case study. Seven municipal buildings of the town of Ādaži, Ādaži Municipality, Latvia, were selected. All buildings are currently 100% energy-dependent: for thermal energy production, natural gas boilers are used, but electricity is purchased from the grid. In addition, all the buildings differ in terms of the total thermal energy and electricity consumption, and also in terms of annual consumption profiles. The ratio between thermal energy and electricity consumption is also different. This allows us to fully evaluate the advantages and disadvantages of chosen technologies and understand the requirements for the efficient use of alternative technologies in each scenario.

This study includes three main alternative scenarios and a baseline or reference scenario for comparison of benefits. Scenarios are divided into two levels depending on the load. Two loads are used—base and reduced. Benchmarks are applied in relation to both thermal energy and electricity consumption. In the case of thermal energy, regression equations are used to describe the correlation between monthly thermal energy demand and outdoor air temperature. All monthly thermal energy consumption that is above the created correlation curve is considered unreasonable, especially in the event of an energy crisis, to ensure the introduction of energy security. All unreasonably high thermal energy consumption above the correlation curve was reduced to the benchmark. Thermal energy consumption below the curve was not changed. In the case of electricity, a benchmark was also applied to reduce consumption.

All scenarios were compared with each other using predefined criteria: (1) 100% self-sufficiency of thermal energy and electricity consumption; (2) specific cost of energy production; (3) total investments and payback period; (4) amount of emissions. Calculations were made for all scenarios, and numerical values were obtained for each criterion. In all alternative scenarios, 100% self-sufficiency was achieved using the biomass boiler and Stirling engine in combination with solar technologies and a heat pump. Dependence on consumer consumption profile and technology characteristics, excess electricity tends to develop. This electricity cannot be used immediately and can, therefore, be sold on the grid.

Further research will explore the feasibility of developing scenarios with the integration of electricity storage technologies.

Specific costs of energy production differ significantly depending on the technologies used. The lowest costs are in scenarios with solar PV and heating pumps. The highest cost for energy production is in the base scenario where natural gas is used. The costs are relatively high in a scenario with the use of a pellet boiler with a Stirling engine. Reducing energy consumption, a slight/small increase in specific costs of energy production can be observed for each of the technologies.

The specific energy cost is significantly influenced by the choice of technologies to be used or expected. The lowest costs are in scenarios that expect the use of solar PV and heat pumps. On the other hand, the highest specific energy cost is in the current situation, where natural gas plays the main role. The proportionally high specific energy cost is in the scenario where only biomass boiler and Stirling engine are expected. By reducing energy consumption, the specific energy cost increases slightly for each technological solution.

The results for total Investments in technology deployment clearly indicate that in Scenarios *S2* and *S3*, the investments are approximately two times higher than in Scenario *S1*. This is because Scenarios *S2* and *S3* use more than one hybrid technology solution. Reducing energy consumption reduces technology-required installation capacity and consequently reduces investments required.

The payback period in each alternative scenario depends on the total investments required and the specific energy cost. Even though specific energy cost for Scenario *S1* is higher than for Scenarios *S2* and *S3*, the payback period is shorter for Scenario *S1*. This is because Scenarios *S2* and *S3* with the use of solar technology have significantly higher total investments.

There are more advanced methods for calculating payback periods [116–118]. Further studies should examine the need for more in-depth economic analysis.

The scenarios considered in this study were also compared based on the amount of emissions produced. Air pollutant emissions of CO, $NO_x$, and PM, as well as greenhouse gas $CO_2$, were calculated. In Scenario *BS*, where a natural gas boiler is used, significantly less CO and $NO_x$ emissions are produced. However, there are significant $CO_2$ emissions. By applying solar technologies in Scenarios *S2* and *S3*, it is possible to reduce air pollutant emissions compared to Scenario *S1*. Reducing energy consumption also reduces the amount of emissions produced in all scenarios.

Lauma Balode et al. have conducted research on the transition from using natural gas to renewable energy sources to producing energy within the boundaries of one village, similar to the case study chosen in this article [119]. The results suggest that the approach of using a heat pump and PV panels in individual heating has a high sustainability index (SI) in comparison to other solutions. In a current study, these technologies are included in scenario *S2*. In the method chosen by Lauma Balode et al., the SI covers all aspects of energy security considered in the current study. When comparing the results related to the emission context, it can be concluded that they are similar—the total emissions of scenario *S2* are the lowest. In the case of the current study, all indicators are broken down separately, which does not reflect the overall view. A similar approach should be followed in further research.

Other RES technologies are suitable for developing energy security. Wind power has the most significant potential. Wind power usage is projected to increase significantly by mid-century [120]. The increase in water energy use is not predicted—it is influenced, among other things, by various national laws restricting the construction of new hydro-electric stations [121], including in Latvia. These two technologies should be included in future studies.

## 6. Conclusions

Creating a sustainable energy supply system for municipal buildings and achieving energy independence is possible. The results also show that before implementing new

energy technologies, it is essential to ensure the reduction of energy consumption. It is crucial to create an energy-efficient, energy-secure thermal and electricity supply.

All alternative Scenarios *S1*, *S2*, and *S3* can reach the requirement for 100% self-sufficient energy consumption and have their advantages and disadvantages as follows:

- Scenario *S1*, where biomass boiler and Stirling engine are expected, is characterized by lower investments and a shorter payback period;
- Scenarios *S2* and *S3*, where also solar technologies are expected, have lower specific energy costs and amount of emissions;
- Increasing energy security involves technological, economic, and environmental aspects;
- Creating a sustainable energy supply system, it is essential to consider all the criteria addressed in this study.

Some factors may affect whether the application of the Stirling engine can be efficient. Important aspects are the ratio of thermal energy and electricity consumption, equalized energy consumption without peaks, and summer thermal energy consumption. The most significant distribution of results between the selected buildings considered is in Scenario *S1*. This concerns both the specific energy cost and the payback period.

The hypothesis was confirmed based on the results obtained. The Stirling engine can achieve 100% energy independence. With an emphasis on environmental aspects (the amount of emissions generated), combining the Stirling engine with solar technologies is the best scenario. At the exact moment, from the economic point of view, solar technology requires increased investments.

This study's main limitation is that an established methodology is for individual energy consumers and not for centralized energy systems. The results reflect a specific situation using technical and economic parameters of the technologies specified in this study.

At the same time, the research made can serve as a basis for other studies where other factors and links between factors that complement the proposed method may be defined. Further studies can also include other energy technologies in new scenarios. It is also possible to have additional criteria for the assessment of scenarios. It is necessary to assess the relevance and importance of each criterion by applying one of the relevant scientific methods, e.g., decision-making analyses.

**Author Contributions:** Conceptualization, J.K. and E.V.; methodology, J.K. and V.K.; validation, V.K., E.V. and D.B.; formal analysis, V.K.; investigation, J.K.; data curation, E.V. and D.B.; writing—original draft preparation, O.S. and A.S.; writing—review and editing, O.S.; visualization, O.S.; supervision, D.B. All authors have read and agreed to the published version of the manuscript.

**Funding:** This research was funded by the Latvian Council of Science, project Alternative biomass knowledge for transition towards energy independence and climate targets (bioenergy Observatory), project No. lzp-2022/1-0414.

**Institutional Review Board Statement:** Not applicable.

**Informed Consent Statement:** Not applicable.

**Data Availability Statement:** The data presented in this study are available on request from the corresponding author.

**Conflicts of Interest:** The authors declare no conflicts of interest.

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
