# Peer review of "Exploring Energy Security and Independence for Small Energy Users: A Latvian Case Study on Unleashing Stirling Engine Potential"

_sustainability, doi:10.3390/su16031224_

Round 1

Reviewer 1 Report

Comments and Suggestions for Authors

The subject of the article is interesting and worth describing. However, the method of implementation requires correction.

The title of the article does not correspond to the content. It is very general, but in the content there is a reference to a specific case.

The abstract does not contain the required elements. Does not correspond to the title and content. There are no specific findings and conclusions from the research.

In the Introduction, the authors presented an introduction to the topic. The Introduction section has deficiencies. There is no clearly defined purpose of the work. The authors actually wrote down two goals of the work. They are different. You need to decide on one main goal (at a more general level) and several specific ones.

There is a research gap missing in the article. It has not been clearly defined.

The Introduction section should include research hypotheses or research questions. At the end there should also be a summary of the content of each section.

The work arrangement requires correction. The content of many section itself is not correct. Subsection titles are missing in the Introduction section. This would make this part more readable. The authors cover many topics.

There is a separate Materials and methods section. However, its content does not fully correspond to the section title. This section should include sources of material and data. Additionally, the methods used must be described. This information should be presented in this order. The research scheme and structure should also be presented. The purposefulness of individual stages of research should also be described. In this part, the authors present the research results. I suggest moving these results (charts) to the research part.

The title of the Case study subsection is misleading. What does it mean? This is a title reserved for the research results section. Additionally, the title of the subsection should be substantive and present relationships and correspond to the content.

Section 3 is titled Results and Discussion. These elements can be combined. However, discussion is missing at the moment. The Discussion section is crucial, the most important in the article. I understand Discussion as referring to other research after presenting your research results. In my opinion, conducting research without a clear comparison and reference to other research means that the obtained results cannot be properly assessed. References to research by other authors should be used.

In the Conclusions section, you should certainly refer to the hypotheses or research questions. Can the hypotheses be verified positively or negatively? Conclusions can be bulleted. Conclusions should be a synthesis. We need to come to specific conclusions. You should not write again about the article's goals and assumptions. There was room for this in section 2. As it stands, the Conclusions section is unacceptable.

Author Response

Greetings,

Thank you for your part in reviewing our manuscript. We appreciate the comments and suggestions you have made. The responses are summarised in the Word file in the attachment.

Respectfully,

M.sc.ing. Oskars Svedovs

Reviewer 2 Report

Comments and Suggestions for Authors

Thank you for allowing me to evaluate this manuscript.

The aim is to explore the possibility of transforming the energy supply process by promoting increased energy security and independence while not reducing energy demand or economic performance.

Please revise the manuscript for styling (e.g., line 39). Use formal and scholarly style.

The introduction is rich in detail, discussing many elements, from sustainable development as a process to direct combustion and solar energy. The whole manuscript is hard to follow as it presents a broader issue (according to its title and premises) and ends up showing the Stirling engine.

The authors must decide on the exact objectives of the study and focus on it. At times, the manuscript seems to be a technical description of various technological installations, which is neither novel nor helpful for the study’s objective. Please revise the manuscript accordingly and present a clean version focused on giving the arguments related to the stated aim for evaluating the scenarios.  

Author Response

Greetings,

Thank you for participating in the review of our manuscript. We appreciate the comments and suggestions made. The attachment contains a Word file with a response.

Respectfully,

M.sc.ing. Oskars Svedovs

Reviewer 3 Report

Comments and Suggestions for Authors

Dear Authors, 

The paper titled "Energy security and independence trilemma: crossroad of technological preconditions economic and environmental aspects" by Janis Kramens et al. presents an extensive and well-structured analysis of energy security and independence, focusing on the town of Ä€daži in Latvia. 

The paper addresses the critical and timely issue of energy security and independence, particularly pertinent given recent global events affecting energy systems​​. The research encompasses a wide range of factors influencing energy security, including political, social, economic, technical, and environmental aspects, providing a holistic view of the issue​​.

However, dispite my positive opinion on this paper, there are several limitations and areas for improvement are identified. 

Remarks:

1.  The study's focus on Ä€daži, Latvia, while detailed, limits the generalizability of its findings. Energy systems and their challenges vary significantly across regions, and the study's insights may not be applicable elsewhere​​.

2. The paper primarily focuses on a few technologies (biomass, solar energy systems, and heat pumps) without adequately exploring or comparing a wider range of potential solutions. This narrow focus could overlook other viable or more efficient technologies​​.

3. While the paper employs a complex methodology, there is a lack of in-depth statistical analysis to validate the scenarios and assumptions made. This raises questions about the reliability and robustness of the conclusions drawn​​.

4. The paper does not sufficiently ground its analysis in existing theoretical frameworks or literature on energy security and sustainability, potentially missing key concepts and debates in the field​​.

5. The paper tends to oversimplify the complex interplay of economic, environmental, and technological factors in energy security. This simplistic approach may lead to misleading or incomplete conclusions​​.

6. The findings and recommendations are highly specific to the case study and may not be easily transferable or applicable to other contexts or larger-scale energy systems​​.

7. The study does not sufficiently address the role of policy and regulatory frameworks in shaping energy security and independence. This omission limits the paper's practical relevance for policymakers​​.

8. The paper does not compare the proposed solutions with other case studies or existing systems in similar settings, which would provide a more balanced and comprehensive understanding of the effectiveness of the proposed scenarios​​.

9. The study lacks a long-term perspective on the sustainability and feasibility of the proposed energy security solutions, particularly in the face of evolving technological and environmental challenges​​.

While the paper tackles an important topic in energy security, its methodological limitations, theoretical gaps, and lack of broader applicability raise significant concerns. To enhance its academic and practical value, the study needs to broaden its scope, deepen its theoretical grounding, and provide a more robust and comprehensive analysis of the complex dynamics of energy security and independence.

Good luck! 

Comments on the Quality of English Language

Minor editing of English language required.

Author Response

Greetings,

Thank you for participating in the review of our manuscript. We appreciate the comments and recommendations made. The attachment contains a Word file with a reply.

Respectfully,

M.sc.ing. Oskars Svedovs

Reviewer 4 Report

Comments and Suggestions for Authors

Please see my comments and try to fix them in most possible ways.

Comments on the Quality of English Language

English of the manuscript is fine to me.

Author Response

(The authors gave the same response as above.)

Round 2

Reviewer 1 Report

Comments and Suggestions for Authors

In response to the review, the authors write that they have improved the article. The last available version still has the old title.

The new proposal is closer to the content, but the title still sounds a bit clumsy.

You must clearly indicate in the text what the research gap is. It must be in the Introduktion section. The reader cannot guess.

Methodology is the study of methods. Do not use this phrase (subsection 2.1.). Correctly it should be Methods.

I still have doubts about the title of section 3 Case study. You can't name a section like that. These are simply research results. Within the research results, you can separate a subsection and name it substantively. Case study is not a substantive title.

Again, there are no references to literature in the Discussion section. How do you want to evaluate your results if you don't compare them with the results obtained by other authors? You don't have to focus exactly on the Stirling engine solution, but also on other relevant aspects. I do not accept the explanation that your topic is so innovative that no one writes about it or the topic discussed.

There is still a lack of synthesis in the Conclusions section. You need to point out the most important conclusions from the article so that the reader has no doubts.

Author Response

Greetings,

Thank you for examining the corrections we made in the first round. We've corrected the manuscript according to the new comments (the new file will be available soon). See our response below. 

Best regards,
M.sc.ing. Oskars Svedovs

Reviewer 4 Report

Comments and Suggestions for Authors

1, Please mention in section 4.5 that 'the study needs depth analysis on the financial key performance indicators (KPIs) to understand the insight of the energy scenarios and to perceive the economic sustainability of the investment'.

2. Please mention in the conclusion section about the limitation for analysis of economic sustainability and the need for future investigation. 

Comments on the Quality of English Language

The English of the manuscript needs minor editing.

Author Response

(The authors gave the same response as above.)

Round 3

Reviewer 1 Report

Comments and Suggestions for Authors

The article has been corrected accordingly.